# Classification of Protein Sequences by a Novel Alignment-Free Method on Bacterial and Virus Families

**DOI:** 10.3390/genes13101744

**Published:** 2022-09-27

**Authors:** Mengcen Guan, Leqi Zhao, Stephen S.-T. Yau

**Affiliations:** 1Department of Mathematical Sciences, Tsinghua University, Beijing 100084, China; 2Yanqi Lake Beijing Institute of Mathematical Sciences and Applications (BIMSA), Huairou District, Beijing 101400, China

**Keywords:** accumulated natural vector, convex hull method, proteins, alignment-free, classification

## Abstract

The classification of protein sequences provides valuable insights into bioinformatics. Most existing methods are based on sequence alignment algorithms, which become time-consuming as the size of the database increases. Therefore, there is a need to develop an improved method for effectively classifying protein sequences. In this paper, we propose a novel accumulated natural vector method to cluster protein sequences at a lower time cost without reducing accuracy. Our method projects each protein sequence as a point in a 250-dimensional space according to its amino acid distribution. Thus, the biological distance between any two proteins can be easily measured by the Euclidean distance between the corresponding points in the 250-dimensional space. The convex hull analysis and classification perform robustly on virus and bacteria datasets, effectively verifying our method.

## 1. Introduction

Proteins, the highly complex substances that are the basic organic matter of life, have been a hot topic for bioinformatics [1]. Proteins play different roles in the processes of life [2], such as enzymes [3]. Proteins carry out the duties specified by gene information. Thus, the research on proteins and protein sequences can reveal the evolutionary relationships of different species. From the molecular biology perspective, a protein sequence is a series of amino acids bonded via peptide bonds, and protein structures vary. There are 20 different types of amino acids that can be combined to make a protein [4]; the sequence of the amino acids determines each protein’s unique three-dimensional structure and specific function. These sequences contain the distribution information of 20 types of amino acids, and a single nucleotide polymorphism (SNP) may result in a change in a protein’s function [5]. There are many reasons for protein diversity [6]. One reason is the diversity of the sequences; proteins are diverse both within and between families, which makes classifying different virus or bacteria families based on protein sequences reliable.

Studying the relationship among different protein sequences is now a matter of great concern in related research. The methods for sequence similarity analysis commonly depend on a multiple sequence alignment, which usually requires a long computation time to obtain results. Therefore, alignment-free methods are proposed to overcome this ineffectiveness. Currently published alignment-free methods include graphical representation [7,8], probabilistic measure [9,10], k-mer [11,12], etc. Furthermore, sequence vector representation methods without alignment are also popular, such as the moment vector [13] and natural vector [14,15].

In this paper, we propose a novel alignment-free method for protein sequences. The accumulated natural vector for genome sequences is a previously published method that performs well on many datasets [16]. However, it ignores the vast field of protein sequences. Similar to the 18-dimensional vector of nucleotide sequences [16], the accumulated natural vector of protein sequences we designed also covers the number, average position, variance, and covariance information of amino acids. Detailed information is introduced in the next chapter. Our method not only considers the basic properties of each amino acid but also the covariance between them. Each protein sequence is in one-to-one correspondence with a point in a 250-dimensional space. Subsequently, the biological distance between two protein sequences is measured by the Euclidean distance between the corresponding points. Therefore, this approach can classify sequences into the correct cluster at a lower time cost without reducing accuracy. We also performed a convex hull analysis, which states that the convex hull formed from the same family’s protein corresponding points do not intersect with other families’ convex hulls. The classification results with high and robust accuracy further validate our methods effectively.

## 2. Materials and Methods

### 2.1. Accumulated Natural Vector for Protein Sequences

Since the distribution of amino acids determines a protein sequence, the use of appropriate models to describe the distribution of amino acids is an important issue. Former discrete models, such as the “pseudo amino acid composition” (PseAAC) model [17], have been applied to the prediction of various protein attributes. The information that can be extracted from the distribution of amino acids is very diverse. Our accumulated natural vector method is a natural description of amino acid distribution information. Assume $S=(s_{1},s_{2},s_{3},\cdots ,s_{N})$ is a protein sequence of length *N*, i.e., $s_{i}\in \left\{A,R,N,D,C,E,Q,G,H,I,L,K,M,F,P,S,T,W,Y,V\right\}$, i=1,2,⋯,N. Each letter represents a type of amino acid, and there are 20 letters representing 20 types of amino acids. For convenience, the set of these 20 letters is denoted as A. This subsection defines the accumulated natural vector of the protein sequence *S*.

#### 2.1.1. Related Definitions

We first define the indicator function of these 20 amino acids, respectively:(1)$$I_{\alpha }\left(i\right)=\left\{\begin{array}{ll} 1 & \mathrm{if}\;s_{i}=\alpha \\ 0 & \mathrm{if}\;s_{i}\neq \alpha \end{array}\right.$$
where α∈A,i=1,2,⋯,N

Next, we define the accumulated indicator function of each amino acid:(2)$${\tilde{I}}_{\alpha }\left(k\right)=\sum_{i=1}^{k}I_{\alpha }\left(i\right)\;\alpha \in A$$

We have defined the indicator function and accumulated indicator function and noticed that there is an obvious property: $n_{\alpha }$, the total amount of the amino acid α in the sequence *S*, is the last column.
(3)$$n_{\alpha }=\sum_{i=1}^{N}I_{\alpha }\left(i\right)={\tilde{I}}_{\alpha }\left(N\right)\;\alpha \in A$$

Now we define the average position of the amino acid α in the sequence *S*:(4)$$\zeta_{\alpha }=\frac{\sum_{i=1}^{N}{\tilde{I}}_{\alpha }\left(i\right)}{n_{\alpha }}\;\alpha \in A$$

For the two different amino acids α and β, define their covariance in the protein sequence *S* as:(5)$$cov\left(\alpha ,\beta \right)=\sum_{i=1}^{N}\frac{\left({\tilde{I}}_{\alpha }\left(i\right)-\theta_{\alpha }\right)\times ({\tilde{I}}_{\beta }\left(i\right)-\theta_{\beta })}{n_{\alpha }\times n_{\beta }}$$
where $\alpha ,\beta \in A,\theta_{\alpha }=\sum_{i=1}^{N}{\tilde{I}}_{\alpha }\left(i\right)/N,\theta_{\beta }=\sum_{i=1}^{N}{\tilde{I}}_{\beta }\left(i\right)/N$. Note that the definition of $\theta_{\alpha }$ here is different from the average position above.

Then, the variance of the amino acid α is the special case of α=β in (Equation 5):(6)$$D_{\alpha }=cov\left(\alpha ,\alpha \right)=\sum_{i=1}^{N}\frac{{\left({\tilde{I}}_{\alpha }\left(i\right)-\theta_{\alpha }\right)}^{2}}{n_{\alpha }^{2}}\;\alpha \in A$$

Thus, we have defined every concept needed in the accumulated natural vector.

#### 2.1.2. Accumulated Natural Vector

Now we can build up the accumulated natural vector of the protein sequence *S*. The first 20 dimensions describe the amount of 20 amino acids, the second 20 dimensions describe the average positions of the 20 amino acids, and the third 20 dimensions describe the variances of the 20 amino acids. The final 202=190 dimensions describe the covariances between each two amino acids. The total number of dimensions is 250.
(7)$$\begin{array}{c} (n_{A},n_{R},\cdots ,n_{V},\zeta_{A},\zeta_{R},\cdots ,\zeta_{V},D_{A},D_{R},\cdots ,D_{V},\\ cov(A,R),cov(A,N),\cdots ,cov(Y,V)) \end{array}$$

For example, take the protein sequence S=(ARRNADCDCC) of length 10. The indicator functions and the accumulated indicator functions are shown in Table 1 and Table 2, respectively. Here, except for A,R,N,D, and *C*, the functions of 15 amino acids are all 0.

According to the tables, we calculate:(8)$$n_{A}={\tilde{I}}_{A}\left(10\right)=2$$
(9)$$\begin{array}{cc} \zeta_{A}= & \frac{\sum_{i=1}^{10}{\tilde{I}}_{A}\left(i\right)}{n_{A}}\\ = & \frac{1+1+1+1+2+2+2+2+2+2}{2}\\ = & 8 \end{array}$$

Similarly, we can obtain $n_{R}=2,n_{N}=1,n_{D}=2,n_{C}=3,\zeta_{R}=8.5,\zeta_{N}=7,\zeta_{D}=4$, and $\zeta_{C}=2.333$.

Next, we calculate the variance and covariance.
(10)$$\begin{array}{cc} \theta_{A}= & \frac{\sum_{i=1}^{10}{\tilde{I}}_{A}\left(i\right)}{10}\\ = & \frac{1+1+1+1+2+2+2+2+2+2}{10}\\ = & 1.6 \end{array}$$
(11)$$\begin{array}{cc} D_{A}= & \sum_{i=1}^{10}\frac{{\left({\tilde{I}}_{A}\left(i\right)-\theta_{A}\right)}^{2}}{n_{A}^{2}}\\ = & \frac{4\times {\left(1-1.6\right)}^{2}+6\times {\left(2-1.6\right)}^{2}}{2^{2}}\\ = & 0.6 \end{array}$$

Similarly, we can obtain $D_{R}=1.025,D_{N}=2.1,D_{D}=1.9$, and $D_{C}=1.122$
(12)$$\begin{array}{cc} cov(A,R)= & \sum_{i=1}^{10}\frac{\left({\tilde{I}}_{A}\left(i\right)-\theta_{A}\right)\times ({\tilde{I}}_{R}\left(i\right)-\theta_{R})}{n_{A}\times n_{R}}\\ = & \frac{1}{2\times 2}(\left(1-1.6\right)\times \left(0-1.7\right)+\\ \; & \left(1-1.6)\times (1-1.7)+(1-1.6)\times (2-1.7\right)\\ \; & \times 2+(2-1.6)\times (2-1.7)\times 6)\\ = & 0.45 \end{array}$$

As well as cov(A,N)=0.9,cov(A,D)=0.8,cov(A,C)=0.467,cov(R,N)=1.05,cov(R,D)=0.6,cov(R,C)=0.35,cov(N,D)=1.2,cov(N,C)=0.7, and cov(D,C)=1.233.

Finally, the accumulated natural vector of *S* is (2,2,1,2,3,0,⋯,0,8,8.5,7,4,2.333,0,⋯,0,0.6,1.025,2.1,1.9,1.122,0,⋯,0,0.45,0.9,0.8,0.467,0,⋯,0,1.05,0.6,0.35,0,⋯,0,1.2,0.7,0,⋯,0,1.233,0,⋯,0).

### 2.2. Convex Hull Method

In the previous section, we introduce how a protein sequence is represented by a 250-dimensional vector. Therefore, the distance between two vectors represents the biological distance of the corresponding two protein sequences. The convex hull principle for protein states that convex hulls corresponding to different families are disjoint with each other [15,18]. The relationship among point sets can be better observed by constructing their convex hulls.

Given a finite point set $A=\{a_{1},a_{2},\cdots ,a_{n}\}$ in $R^{k}$ space, we define the convex hull of *A* as:(13)$$C\left(A\right)=\left\{p|p=\sum_{i=1}^{n}\lambda_{i}a_{i},\sum_{i=1}^{n}\lambda_{i}=1,\lambda_{i}\geq 0,1\leq i\leq n\right\}$$

Generally speaking, the convex hull of a given point set is the smallest convex set that contains this point set.

There are many ways to judge whether two convex hulls intersect. Considering that the construction of a convex hull in a high-dimensional space is computationally intensive and time-consuming, we do not directly consider two convex hulls here. Instead, we consider two point sets to determine whether the convex hull formed by them intersects.

Given two finite point sets in the $R^{k}$ spaces $A=\{a_{1},a_{2},\cdots ,a_{n}\}$ and $B=\{b_{1},b_{2},\cdots ,b_{m}\}$, we want to determine whether the convex hulls of A and B intersect. From the definition given in (Equation 13), if some sets of coefficients satisfy:(14)$$\begin{array}{c} \sum_{i=1}^{n}\lambda_{i}a_{i}=\sum_{j=1}^{m}\mu_{j}b_{j},\\ \sum_{i=1}^{n}\lambda_{i}=1,\\ \sum_{j=1}^{m}\mu_{j}=1,\\ 0\leq \lambda_{i},\mu_{j}\leq 1,\\ 1\leq i\leq n,1\leq j\leq m, \end{array}$$

Then the two convex hulls of A and B intersect; otherwise, the two convex hulls are disjoint [18].

### 2.3. Convex Hull Distance

The distance between the two sequences we have defined is the Euclidean distance between their corresponding natural vectors. Then, the distance between two convex hulls corresponding to the two families is the Euclidean distance between the centers of the two convex hulls constructed by these natural vectors.
(15)distance=∥meanvector(A)−meanvector(B)∥

*A*, *B* are the two sets of 250-dimensional accumulated natural vectors of the two families. Here, *meanvector*(*A*) represents the center of the convex hull composed of set *A*. The expression of *meanvector*(*A*) is:(16)$$meanvector\left(A\right)=\frac{1}{n}\sum_{i=1}^{n}A_{i}$$
where *n* is the number of vectors in set *A*, $A_{i}$ is a vector in set *A*. With this convex hull distance definition, it is easy to calculate the distance of two convex hulls if we know their point sets.

### 2.4. Linear Discriminant Analysis

With the help of the accumulated natural vector and the convex hull method, we can construct convex hulls and study their intersection in a 250-dimensional space. When we want to visualize convex hulls and research their properties in a lower-dimensional space, especially in a 2-dimensional space, the dimensional reduction method in cluster analysis is indispensable. First, we introduce linear discriminant analysis, which is the widely known method.

We define $X_{j}\left(j=0,1\right)$ as the set of samples in Class *j*. In our experiment, the two sets of samples are two point sets of the convex hulls whose intersection is our study subject.

Our goal is to find the projection direction ω that maximizes the sample distance between the two projected categories (0,1).

We define $N_{j}\left(j=0,1\right)$ as the number of samples in Class *j*, $\mu_{j}\left(j=0,1\right)$ as the mean value of the samples in Class *j*, and $\sum_{j}$ as the covariance matrix of the samples in Class *j*. $\mu_{j}\left(j=0,1\right)$ can be calculated by:(17)$$\mu_{j}=\frac{1}{N_{j}}\sum_{x\in X_{j}}x\left(j=0,1\right)$$

$\sum_{j}$ can be calculated by:(18)$$\sum_{j}=\sum_{x\in X_{j}}\left(x-\mu_{j}\right){\left(x-\mu_{j}\right)}^{T}\left(j=0,1\right)$$

As previously mentioned, the LDA (short for linear discriminant analysis) algorithm should make two projected classes of datasets ($X_{1},X_{2}$) in the projection direction ω as far as possible, so we need to maximize ${∥\omega^{T}\mu_{0}-\omega^{T}\mu_{1}∥}_{2}^{2}$ (using the Euclidean distance here) and minimize the within-group covariance in the meantime.

The within-class scatter matrix $S_{\omega }$ is defined as:(19)$$S_{\omega }=\sum_{0}+\sum_{1}=\sum_{x\in X_{0}}\left(x-\mu_{0}\right){\left(x-\mu_{0}\right)}^{T}=\sum_{x\in X_{1}}\left(x-\mu_{1}\right){\left(x-\mu_{1}\right)}^{T}$$

The between-class scatter matrix $S_{b}$ is defined as:(20)$$S_{b}=\left(\mu_{0}-\mu_{1}\right){\left(\mu_{0}-\mu_{1}\right)}^{T}$$

Then, the two-class LDA can be formulated as an optimization problem to find a set of linear combinations with the coefficient ω.
(21)$$argmaxJ\left(\omega \right)=\frac{\omega^{T}S_{b}\omega }{\omega^{T}S_{\omega }\omega }$$

The expression is the Rayleigh Quotient R(A,x). The solution ω to the above optimization function J(ω) is the eigenvector corresponding to the maximum eigenvalue of the matrix $S_{\omega }^{-1}S_{b}$. In the case of projecting to two dimensions, we need to calculate the two eigenvectors that correspond to the two largest eigenvalues of the matrix $S_{\omega }^{-1}S_{b}$. This step can be accomplished by MATLAB programming or another program.

### 2.5. Maximum Margin Criterion

When we use LDA to complete the dimensionality reduction, we may meet the small sample size problem [19], which is caused by the singular $S_{\omega }$. The maximum margin criterion (MMC) is proposed to avoid the small sample size problem and calculate the most discriminant vectors. The MMC method is simple, efficient, and stable compared to the PCA+LDA method [20].

Consider a linear mapping $W\in R^{D\times d}$, where D and d are the dimensionalities of the data before and after the projection. The MMC introduces a new objective function:(22)$$J\left(\omega \right)=tr(\omega^{T}\left(S_{b}-S_{\omega }\right)\omega )$$

We can suppose that $\omega^{T}\omega =1$ because ω can be multiplied by any constant to make it be the unit vector. Then, we just need to solve the following constrained optimization:(23)$$\begin{array}{c} \sum_{k=1}^{d}\omega^{T}\left(S_{b}-S_{\omega }\right)\omega , \end{array}$$(24)$$\begin{array}{c} subject\;to\;\omega_{k}^{T}\omega_{k}=1,k=1,\cdots ,d \end{array}$$

The small sample size problem is avoided by the formation $S_{b}-S_{\omega }$ instead of $S_{\omega }^{-1}S_{b}$ of the LDA.

### 2.6. Knn Classification

The classification performance of the 250-dimensional accumulated natural vector is also required in this paper. The K-nearest neighbor is a simple classification method and has been developed successfully in real applications [21]. The choice of the *k*-value has a huge impact on the final classification results. In practical applications, cross-checking is usually required to select the optimal *k*-value [22]. In particular, we achieved good results when we chose k=1 in our work.

Suppose the training set contains *N* samples $\left\{x_{1},x_{2},\cdots ,x_{N}\right\}$, which belong to *t* classes $\left\{A_{1},A_{2},\cdots ,A_{t}\right\}$. Define the Euclidean distance from the unclassified sample *x* to the training set sample $x_{i}$ as $d(x,x_{i})$, if:
$$d\left(x,x_{k}\right)=min_{i=1,2,\cdots ,N}\left(d\left(x,x_{i}\right)\right),x_{k}\in A_{j}$$

Then, the nearest neighbor classification decision is: $x\in A_{j}$.

## 3. Results and Discussion

### 3.1. Convex Hull Analysis of Bacterial Families

Bacteria are almost everywhere on earth and play an important role in many research fields [23]. The number of bacteria-related protein sequences is enormous and still rising. As of May 2020, the Reference Sequence Database (RefSeq) on NCBI had a total of 140 million bacterial protein sequences. Therefore, we selected 117 bacterial families from 13 different phyla and 150 protein sequences per these families for a total of 117×150=17,550 protein sequences. Appendix A provides the complete list of the 117 bacterial families.

For each protein sequence, we calculated the 250-dimensional accumulated natural vector. Thus, each protein sequence corresponded to a point in a 250-dimensional space, and 117 different bacteria families corresponded to 117 finite point sets in a 250-dimensional space. We considered the convex hull of these 117 finite point sets and verified whether they intersect in pairs. There were 1172=6786 pairs of convex hulls. No intersection was observed using the method in Section 2.2. We were not surprised because the convex hull method had similar results when applied to other datasets in previous work [15]. We applied the linear discriminant analysis method to visualize the results. For example, Figure 1 shows that the Acetobacteraceae family disjoints the Acidiferrobacteraceae family.

This result shows that our proposed 250-dimensional accumulated natural vector is very effective when applied to protein sequences. The disjointness property reveals that our method accurately clusters protein sequences from the same family. It is worth noting that the advantage of using the convex hull method is that even if two points are closer together, it does not prevent them from being in different convex hulls. We can suppose that the convex hulls of the different bacterial families are pairwise disjoint. For a new, unclassified protein sequence, we can calculate its 250-dimensional accumulated natural vector in the same way and then analyze in which convex hull the point is located. Subsequently, this new protein is classified into the corresponding family. This is the most intuitive application of our method.

### 3.2. Classification of Protein Enzyme Classes

We performed a classification analysis on protein enzyme classes using the accumulated natural vector. Four datasets were selected from UniProt. According to UniProt, each dataset is composed of seven enzyme classes: Oxidoreductases, Transferases, Hydrolases, Lyases, Isomerases, Ligases, and Translocases. We used the one-nearest neighbor classification method mentioned in Section 2.6 to divide these protein sequences into the seven enzyme classes. All protein sequences appeared in the training set and the test set. By predicting the enzyme classes in the test set and comparing these with the training set, we calculated the accuracy of our classification method.

*E. coli* (*Escherichia coli*) is a type of bacteria closely related to human life [24]. The *E. coli* dataset contained 12284 protein sequences. A total of 11788 protein sequences are classified accurately, and the total accuracy is 11,782/12,284(95.9%). The detailed results of this classification are shown in Table 3.

This method was also applied to the other three bacterial family datasets. The total accuracies were 6652/6890(96.5%), 3940/4017(98.1%), and 4511/4655(96.7%), respectively. Appendix A show the detailed results of classification.

Using the one-nearest neighbor classification method, one sequence is incorporated into its nearest convex hull, which represents one bacterial family. The high accuracy rate results indicate that our method combined with the one-nearest neighbor algorithm performed robustly in the classification of protein enzyme classes. Although the accuracy cannot reach 100%, it is still a good result. We must admit that our dataset does not guarantee that every point with the nearest distance is put into the correct convex hull, which is equivalent to saying that two points that are close together can be in different convex hulls. This non-ideal situation is possible but at a low frequency. This is also another piece of evidence to support the notion that accumulated natural vectors provide a good representation of protein space.

### 3.3. Convex Hull Analysis of Virus Families

#### 3.3.1. Intersection of Virus Families in a 250-Dimensional Space

We chose 73 virus families and downloaded all of their reviewed protein sequences. Detailed information about the dataset is shown in Appendix A. Then, we obtained the accumulated natural vector of each protein sequence in a 250-dimensional space. These vectors of different families constituted the different vector sets. We wanted to determine whether the convex hulls of these vectors intersected in 250 dimensions. Here, we introduce the Baltimore virus taxonomy, which divides viruses into seven categories based on differences in the gene expressions of different viruses.

Our expected result was that all virus families under the same Baltimore virus category would be disjoint.

For one Baltimore classification, we tested the intersection of the convex hulls under each Baltimore classification. The test method follows the theorem in Section 2.2; the input was the two “point set” (vector set), and the output was the intersection of the convex hull pairs.

The intersection results of the 73 virus families are shown in Table 4. A total of 738 pairs of convex hull pairs were counted. Among these, 727 pairs were disjoint in 250 dimensions, and 11 pairs intersected in 250 dimensions. The disjoint convex hull pairs accounted for 0.9851 of all convex hull pairs.

Among the seven Baltimore taxonomies, several convex hulls intersect under the dsDNA Baltimore taxonomy, and all convex hulls disjoint under the other six Baltimore taxonomies. There are 11 intersecting dsDNA virus family pairs in the 250-dimensional space, as shown in Appendix A.

Then, we used the distances of the convex hulls corresponding to these 11 virus family pairs to perform a cluster analysis. The convex hull distances calculated here were the Euclidean distance between the two centers of the two convex hulls defined in Section 2.3. The result of our cluster analysis is in Figure 2.

From Figure 2, we can see that the three nearest virus families are *Podoviridae, Siphoviridae and Myoviridae*, which all belong to *Caudovirales* and are parasitic in bacteria. Their convex hulls have closer distances, so we deduce that these three families have closer evolutionary distances.

Similarly, we performed a cluster analysis of all virus families in the dataset by calculating the distance matrix of their convex hulls. The result is that all virus families are divided into three clusters. Cluster 2 includes *Togaviridae, Tobaniviridae, Secoviridae, Potyviridae, Picornaviridae and Hypoviridae*. Cluster 3 includes *Iflaviridae and Flaviviridae*. To visualize the result of the cluster analysis, we represent each virus family by the mean vector of its corresponding convex hull, and we project these points onto a 2-dimensional plane. The result is shown in Figure 3.

As shown in Figure 3, we can see that all virus families are divided into three clusters in a 2-dimensional space, and the virus families in each cluster remain consistent with those in a 250-dimensional space.

Above all, most of the convex hull pairs of the 73 virus families are disjoint in a 250-dimensional space. Those virus families whose convex hulls intersect with those of the other families are all from the dsDNA Baltimore taxonomy. Although these virus families have intersecting convex hulls with each other, their evolution relationships can also be reflected by the convex hull distances.

#### 3.3.2. Intersection of Virus Families in a 2-Dimensional Space

Visualization of the results of classification is very important. Convex hulls in a 2-dimensional space are much more intuitive than those in 250-dimensional space. Thus, we project the convex hulls constructed above onto 2-dimensional space to observe their intersection. The method used is the maximum margin criterion, instead of the traditional method of linear discriminant analysis [25], to avoid the small sample size problem [22]. A part of the results of the classification is shown in Table 5, and the complete result is shown in Appendix A. Compared to our newly proposed 250-dimensional natural vector of protein sequences, the intersection result of a 60-dimensional natural vector [26] is bad in a 2-dimensional space. All convex hulls intersect in a 2-dimensional space. The detailed results are shown in Appendix A. This illustrates that the introduction of covariance between the amino acids is an improvement for the intersection analysis.

The percentage of non-intersecting convex hull pairs is nearly 0.85 for most of the virus families, except for *Adenoviridae*, *Herpesviridae*, *Iridoviridae*, *Mimiviridae*, *Myoviridae*, *Podoviridae*, *Poxviridae*, *Reoviridae*, and *Siphoviridae*.

We dropped the above nine families whose convex hulls intersect with the convex hulls of other families in a 250-dimensional space, and then the percentage of non-intersecting convex hull pairs is 0.9588.

In conclusion, most of the convex hulls of the virus families disjoint both in a 250-dimensional space and a 2-dimensional space. The percentage of non-intersection of the virus families may be improved by incorporating other elements in the accumulated natural vector; for example, we can incorporate higher-order moments of the 20 amino acids. Correspondingly, the addition of more elements will add to the dimensions of the vectors and make calculations more complicated.

## 4. Conclusions

In this paper, the 250-dimensional accumulated natural vector method is proposed by describing the distribution of 20 amino acids within a protein sequence. Protein sequences with similar properties correspond to closer points, so proteins in the same family tend to cluster together. Our proposed method makes it easy to classify the protein sequences since it avoids the high computational complexity associated with sequence alignment and takes advantage of mathematical concepts only. Its applications to real datasets suggest that the accumulated natural vector method is a powerful tool for the classification of protein sequences. However, there is still a lot of room for improvement of this novel method by incorporating other elements. 

## Figures and Tables

**Figure 1 genes-13-01744-f001:**
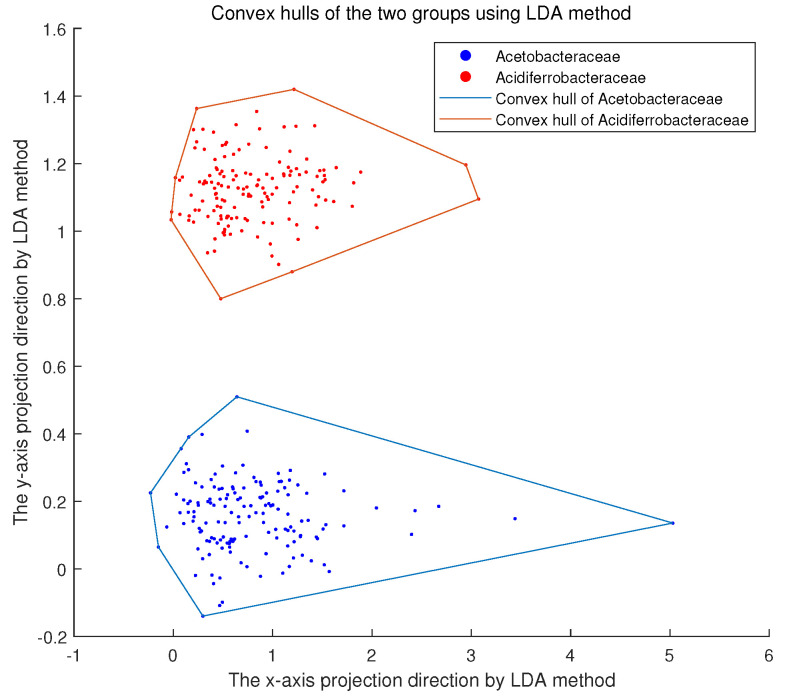
Convex hulls of bacterial family Acetobacteraceae and bacterial family Acidiferrobacteraceae after dimension reduction by LDA(Linear Discriminant Analysis) method.

**Figure 2 genes-13-01744-f002:**
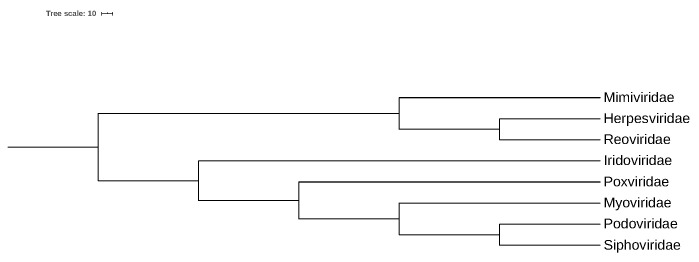
Phylogenetic tree of eight virus families by NJ method: *Reoviridae*, *Herpesviridae*, *Mimiviridae*, *Iridoviridae*, *Poxviridae*, *Myoviridae*, *Siphoviridae*, *Podoviridae*.

**Figure 3 genes-13-01744-f003:**
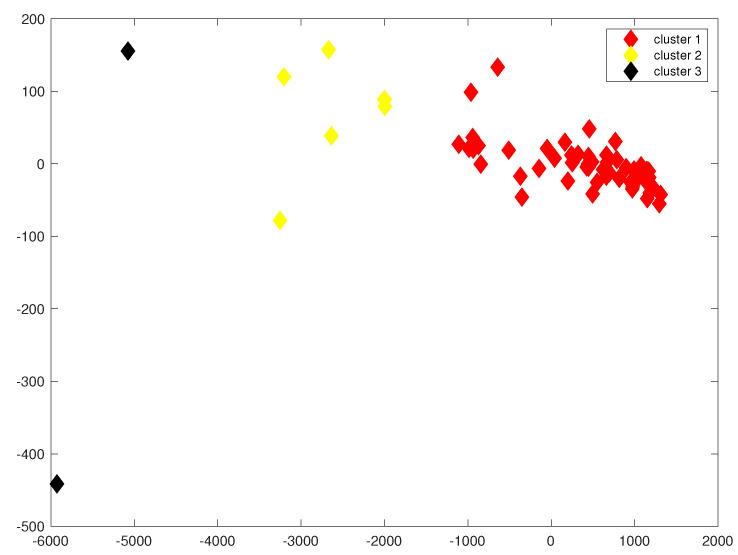
Cluster analysis of all virus families by shortest distance method: red, yellow and black points represent different virus families in three clusters. *x*-axis and *y*-axis are the two coordinate axes of the two-dimensional plane. Values on the axes are position coordinates.

**Table 1 genes-13-01744-t001:** The indicator functions of S.

Sequence S	A	R	R	N	A	D	C	D	C	C
Position(i)	1	2	3	4	5	6	7	8	9	10
$I_{A}\left(i\right)$	1	0	0	0	1	0	0	0	0	0
$I_{R}\left(i\right)$	0	1	1	0	0	0	0	0	0	0
$I_{N}\left(i\right)$	0	0	0	1	0	0	0	0	0	0
$I_{D}\left(i\right)$	0	0	0	0	0	1	0	1	0	0
$I_{C}\left(i\right)$	0	0	0	0	0	0	1	0	1	1

**Table 2 genes-13-01744-t002:** The accumulated indicator functions of S.

Sequence S	A	R	R	N	A	D	C	D	C	C
Position(i)	1	2	3	4	5	6	7	8	9	10
${\tilde{I}}_{A}\left(i\right)$	1	1	1	1	2	2	2	2	2	2
${\tilde{I}}_{R}\left(i\right)$	0	1	2	2	2	2	2	2	2	2
${\tilde{I}}_{N}\left(i\right)$	0	0	0	1	1	1	1	1	1	1
${\tilde{I}}_{D}\left(i\right)$	0	0	0	0	0	1	1	2	2	2
${\tilde{I}}_{C}\left(i\right)$	0	0	0	0	0	0	1	1	2	3

**Table 3 genes-13-01744-t003:** Classification results of the *E. coli* dataset.

Enzyme Class	Total Number	Correct Number	Accuracy
Oxidoreductases	1829	1720	0.940404
Transferases	4054	3905	0.963246
Hydrolases	2783	2632	0.945742
Lyases	1387	1354	0.976208
Isomerases	823	788	0.957473
Ligases	906	893	0.985651
Translocases	502	490	0.976096
Total	12,284	11,782	0.959134

**Table 4 genes-13-01744-t004:** Baltimore virus taxonomy.

Virus Classification	Samples	Number of Virus Families
dsDNA virus	Herelleviridae	23
ssDNA virus	Microviridae	7
dsRNA virus	Totiviridae	8
ssRNA(+) virus	Alphatetraviridae	30
ssRNA(-) virus	Bornaviridae	1
ssRNA-RT virus	Metaviridae	1
dsDNA-RT virus	Caulimoviridae	2

**Table 5 genes-13-01744-t005:** Partial results of the convex hull intersection: ‘Percentage’ in this Table is the percentage of disjoint convex hull pairs of all convex hull pairs within one virus family of non-intersection.

Virus Family	Percentage	Virus Family	Percentage
Adenoviridae	0.6667	Hepeviridae	0.9861
Alloherpesviridae	0.8472	Herelleviridae	0.8472
Alphaflexiviridae	0.8056	Herpesviridae	0.1389
Alphatetraviridae	1.0000	Hypoviridae	1.0000
Ampullaviridae	0.8472	Inoviridae	0.8056
Anelloviridae	0.9861	Iridoviridae	0.5333
Arteriviridae	0.9583	Kitaviridae	0.9861
Ascoviridae	0.9444	Lavidaviridae	0.9583
Astroviridae	1.0000	Leviviridae	0.9444

## Data Availability

All datasets used in this study may be found here: https://github.com/Gmcen20/ANV250 (accessed on 19 September 2022) and http://www.uniprot.org/ (accessed on 19 September 2022).

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
