# Peer review of "Classification of Protein Sequences by a Novel Alignment-Free Method on Bacterial and Virus Families"

_genes, 2022, doi:10.3390/genes13101744_

Round 1

Reviewer 1 Report

This is an interesting work, but I have some major comments at fundamentals, which are as follows. 

1. A 250-dimensional vector representation of a protein sequence is proposed in the paper, which is the main contribution of the paper. However, it is not clear the motivations of the authors behind choosing these particular 250 numbers. Formulas for each numbers are given, but their motivations, practical implications and characteristics are missing. Author must describe, with appropriate supporting references/example/arguments, why these particular 250 numbers are chosen and why they are expected to explain the biological characteristics of the sequence.

2. In continuation of the previous point, why all 250 numbers are necessary for classification of the protein sequence?  Can some smaller vector be considered to give similar classification?

Since author has applied some dimension reduction technique at a later stage of analyses, there must be some unnecessary variables. Can the author give more insights on their dimension reduction results to suggest which of the 250 numbers can be ignored for the classification purpose.

3. Authors have postulated that "protein sequences from the same familly are likely clustered, rather than being randomly distributed". Is there any biological justification for such postulates? Is there any existing literature to support this postulate? 

4. Are the results expected to be valid for protein sequence other than bacterial, enzyme and virus families? Why do the authors expect so?

5. Are the results replicable? As this is a very important issue, I suggest that the author must submit all their codes and data used in the analyses, either as Supplementary material or through some open source software/repository (e.g., GitHub), so that we can verify the replicability of all the results. 

Reviewer 2 Report

Minor issues:

1. there are some unnecessary breaks in several equations, l47, l82,..., l85, ...

2. It does not seem clear whether statistical software was used in the LDA and Cluster analysis. If any software was used, it is only fair that it be referenced.

Main issues:

1. In this type of investigation where the results are obtained empirically, it seems necessary for the authors to justify that the method produces better results than other validated methods. In my opinion, it will be necessary to compare this new methodology with another that has already been validated. It would also be an asset if the results were obtained in several independent samples so that the variability of the percentage of accuracy could be analyzed.

2. It is also suggested that the authors justify the quality of the accuracy percentages. For research purposes, for example, is an accuracy of 98% a good indicator?

Round 2

Reviewer 1 Report

I am not satisfied with the responses of the authors to my major comments number 1-3 in earlier round of revision. Authors have tried to bypass answering the main questions raised in these comments just by coting one reference, which I find to be an extremely lazy attitude to prepare the revision. My main concerns are still unanswered in this revision.  

Additionally, the present version contains too many ?? sings, where the actual citations/figure numbers, as well as the list of references (including the new ones that the author mentioned in response letter) are missing. Another thing illustrating authors' carelessness! 
